# Osteosarcoma-Induced Pain Is Mediated by Glial Cell Activation in the Spinal Dorsal Horn, but Not Capsaicin-Sensitive Nociceptive Neurons: A Complex Functional and Morphological Characterization in Mice

**DOI:** 10.3390/cancers16101788

**Published:** 2024-05-07

**Authors:** Noémi Bencze, Bálint Scheich, Éva Szőke, Imola Wilhelm, Sándor Körmöndi, Bálint Botz, Zsuzsanna Helyes

**Affiliations:** 1Department of Pharmacology and Pharmacotherapy, Medical School, University of Pécs, 7624 Pécs, Hungary; bencze.noemi89@gmail.com (N.B.); eva.szoke@aok.pte.hu (É.S.); botz.balint@pte.hu (B.B.); 2National Laboratory for Drug Research and Development, Magyar Tudósok Krt. 2, 1117 Budapest, Hungary; 3Department of Pathology and Experimental Cancer Research, Faculty of Medicine, Semmelweis University, Üllői út 26, 1085 Budapest, Hungary; scheich.balint@semmelweis.hu; 4Hungarian Research Network, Chronic Pain Research Group (HUN-REN-PTE), 7624 Pécs, Hungary; 5Institute of Biophysics, HUN-REN Biological Research Centre, 6726 Szeged, Hungary; wilhelm.imola@brc.hu; 6Department of Traumatology, Faculty of Medicine, University of Szeged, 6720 Szeged, Hungary; kormondi.sandor.pal@med.u-szeged.hu; 7Department of Medical Imaging, Medical School, University of Pécs, 7624 Pécs, Hungary; 8PharmInVivo Ltd., Szondy György Str. 10, 7629 Pécs, Hungary

**Keywords:** bone, tumor, microglia, astrocyte, resiniferatoxin, chronic pain, hyperalgesia

## Abstract

**Simple Summary:**

Bone cancer-related chronic pain is mediated by central sensitization involving neuroinflammation via glial cell activation in the spinal dorsal horn, but not the capsaicin-sensitive sensory neuronal system.

**Abstract:**

Bone cancer and its related chronic pain are huge clinical problems since the available drugs are often ineffective or cannot be used long term due to a broad range of side effects. The mechanisms, mediators and targets need to be identified to determine potential novel therapies. Here, we characterize a mouse bone cancer model induced by intratibial injection of K7M2 osteosarcoma cells using an integrative approach and investigate the role of capsaicin-sensitive peptidergic sensory nerves. The mechanical pain threshold was assessed by dynamic plantar aesthesiometry, limb loading by dynamic weight bearing, spontaneous pain-related behaviors via observation, knee diameter with a digital caliper, and structural changes by micro-CT and glia cell activation by immunohistochemistry in BALB/c mice of both sexes. Capsaicin-sensitive peptidergic sensory neurons were defunctionalized by systemic pretreatment with a high dose of the transient receptor potential vanilloid 1 (TRPV1) agonist resiniferatoxin (RTX). During the 14- and 28-day experiments, weight bearing on the affected limb and the paw mechanonociceptive thresholds significantly decreased, demonstrating secondary mechanical hyperalgesia. Signs of spontaneous pain and osteoplastic bone remodeling were detected both in male and female mice without any sex differences. Microglia activation was shown by the increased ionized calcium-binding adapter molecule 1 (Iba1) immunopositivity on day 14 and astrocyte activation by the enhanced glial fibrillary acidic protein (GFAP)-positive cell density on day 28 in the ipsilateral spinal dorsal horn. Interestingly, defunctionalization of the capsaicin-sensitive afferents representing approximately 2/3 of the nociceptive fibers did not alter any functional parameters. Here, we provide the first complex functional and morphological characterization of the K7M2 mouse osteosarcoma model. Bone-cancer-related chronic pain and hyperalgesia are likely to be mediated by central sensitization involving neuroinflammation via glial cell activation in the spinal dorsal horn, but not the capsaicin-sensitive sensory neuronal system.

## 1. Introduction

Osteosarcoma is one of the most common primary malignant tumors of the bone [1]. More than 3.4 million people are affected worldwide every year, with a bimodal age distribution around 16–18 and 60–65 years, and is slightly more common in males [1]. Survival rates can vary widely depending on various factors, from less than 25% to 50–60% [2,3]. In children, teenagers, and young adults, osteosarcoma usually starts in areas where the bone grows quickly, such as near the epiphyseal plates, but can present in any bone in the body, most commonly around the knee and the proximal humerus [4]. Bone tumors can be purely lytic, blastic or a combination of the two [5].

The current medications used to manage bone cancer pain are not considered effective or safe enough to provide adequate pain relief. The goal of the present strategy is to reduce the tumor size by neoadjuvant chemotherapy to achieve complete remission of the disease [6].

Cancer-induced bone pain (CIBP) is caused by the primary bone tumor or metastases originating from breast, prostate and lung cancers [7]. It is usually a severe, persistent, spontaneous and complex pain, often accompanied by hyperalgesia [8]. CIBP is a huge unmet medical need, since the currently used opioid analgesics often have insufficient therapeutic efficacies, exert severe side effects, and induce tolerance and dependence. Unravelling the molecular mechanisms of bone cancer and its related pain development and progression is important to determine novel therapeutic targets [9].

The role of nerves in the development of malignancies has become recognized as a significant aspect of the tumor microenvironment [10]. Recent evidence suggests that cancers may reactivate nerve-dependent developmental and regenerative processes to enhance their growth and survival. Furthermore, sensory, sympathetic and parasympathetic fibers interact with the tumor and stromal cells to initiate and promote the progression of a variety of solid and hematological malignancies [11].

Transient receptor potential (TRP) non-selective cation channels are crucial receptors in the transduction of nociceptive stimuli. They can be activated by a range of molecules produced by the tumor microenvironment through predominantly Ca^2+^-dependent regulation of intracellular signaling pathways. This leads to the peripheral sensitization of the primary nociceptive neurons, resulting in hyperalgesia and allodynia. TRP vanilloid 1 (TRPV1) and ankyrin 1 (TRPA1) abundantly expressed on capsaicin-sensitive peptidergic sensory fibers have been suggested to be involved in chronic tumoral pain, including bone cancer [12]. The periosteum is densely innervated by these peptidergic nociceptive fibers being activated and sensitized by a broad range of mediators, such as bradykinin, endothelin, epidermal growth factor, glial cell line-derived neurotrophic factor, histamine, nerve growth factor, prostaglandins, tumor necrosis factor and vascular endothelial growth factor produced by tumor and immune cells [13,14]. These fibers are selectively activated by capsaicin, the pungent alkaloid of hot peppers and its ultra-potent analogue resiniferatoxin (RTX) through the TRPV1 calcium-permeable non-selective cation channels [15]. Continuous or repeated TRPV1 activation induces prolonged Ca^2+^ influx resulting in defunctionalization of the nerve terminals towards all stimuli, which is a commonly used experimental tool to study the role of capsaicin-sensitive fibers in different pathophysiological processes [16,17].

Interactions between central glial cells (astrocytes and microglia) and neurons in the pain circuit are also critical contributors to the pathogenesis of chronic pain. Both glial cells contribute to neuroinflammation by producing a range of inflammatory mediators (i.e., cytokines, chemokines, growth factors, etc.) and consequent neuronal activation or sensitization in the central nervous system (CNS) [18,19]. Microgliosis and astrogliosis are nonspecific reactive changes manifested in their proliferation, hypertrophy or ramification with increased expression of the activation markers, ionized calcium-binding adapter molecule (Iba1) [20] and glial fibrillary acidic protein (GFAP), respectively [21,22]. They can rapidly modulate synaptic plasticity and subsequently prolong and aggravate pain after tissue and nerve injury [23].

These mechanism are involved in the maintenance of chronic pain, while the number and dysfunction of these cells in the spinal dorsal horn were described in several models of cancer-related pain [24]. Astrocytes were shown to be activated in various rodent chronic pain models in the spinal cord and brain areas involved in processing the sensory and affective components of pain, including the anterior cingulate cortex, somatosensory cortex (SSC), amygdala, thalamus and ventrolateral periaqueductal gray matter [25,26,27,28]. Several studies suggested the role of activated microglial cells in the dorsal horn of the spinal cord that is associated with chronic pain states [20,29,30]. Since the peripheral and central sensitization mechanisms of CIBP are unclear, here, we performed a complex functional and morphological characterization of an osteosarcoma mouse model to investigate the role of capsaicin-sensitive peptidergic nerves. Based on well-established differences in sex-dependent pain-sensing and processing mechanisms both in humans and animal models, male and female mice were investigated in our studies [31,32]. This study represents the first comprehensive examination of the K7M2 orthotopic osteosarcoma mouse model, focusing particularly on pain using in vivo functional and imaging methods. It tracks the tumor’s progression alongside measures of spontaneous and referred pain, as well as neuroinflammation in the spinal dorsal horn, potentially elucidating central pain sensitization mechanisms. Interestingly, the findings suggest that the capsaicin-sensitive subset of nociceptive neurons may not play an explicit role in osteosarcoma-related pain behaviors and tumor growth.

## 2. Materials and Methods

### 2.1. Experimental Animals

We utilized 32 9–10-week-old male and female BALB/c mice, which were bred and housed at the Laboratory Animal House of the Department of Pharmacology and Pharmacotherapy of the University of Pécs. The mice were maintained at a temperature of 24–25 °C and provided with standard rodent chow and water ad libitum, following a 12 h dark/12 h light cycle. This study adhered to European legislation (directive 2010/63/EU) and Hungarian Government regulations (40/2013., II. 14) regarding the protection of animals used for scientific purposes. Approval for the project was obtained from the Animal Welfare Committee of the University of Pécs and the National Scientific Ethical Committee on Animal Experimentation of Hungary. Additionally, this study was licensed by the Government Office of Baranya County (license No. BA02/2000-32/2018). Every effort was made to minimize the number of experimental animals involved in this study. The general health status and body weight measurements of the animals were observed every other day. The experimental protocol with the investigational techniques is summarized in Figure 1.

### 2.2. Osteosarcoma Cell Line

The K7M2 osteosarcoma cell line was provided by Dennis Klinman, NCI-Frederick, MD, USA, and cultured in ATCC-formulated Dulbecco’s Modified Eagle’s Medium (DMEM Thermo Fisher Scientific, Waltham, MA, USA) supplemented with 2 mmol L-glutamine, 10% fetal bovine serum (FBS, Thermo Fisher Scientific), and 1% penicillin and streptomycin (Life Technologies, Carlsbad, CA, USA) to obtain 75% confluency. Vented tissue culture flasks (Nunc Cell Culture Systems, Rochester, NY, USA) were maintained at 37 °C in a humidified atmosphere of 5% carbon dioxide in air, and the cells were split when they reached approximately 80% to 90% confluence. The cell density was adjusted to 5 × 10^5^ in 10 μL PBS for inoculation into the mouse tibia [33,34].

### 2.3. Sensory Desensitization by RTX Pretreatment

The inclusion of capsaicin-sensitive neurons in tumor progression and related pain behaviors was examined by inducing long-lasting desensitization using systemic pretreatment with RTX. RTX was administered subcutaneously at doses of 10, 20, 70 and 100 µg/kg over four consecutive days, two weeks prior to tumor inoculation. Additionally, the mice received a solution containing 4% terbutaline sulfate, 4% theophylline-ethylene diamine and 2% atropine sulfate to mitigate systemic, primarily respiratory, side effects of the RTX treatment. The success of sensory defunctionalization was confirmed two weeks later, before K7M2 injection, by the absence of eye-wiping behavior following a 10 µL 0.1% capsaicin drop [35,36].

### 2.4. Intratibial Injection of K7M2 Cells

The mice were deeply anesthetized by intraperitoneal injection of Na-Pentobarbital (Euthanimal, Alfasan Netherland, BV). One hind limb was shaved, and antisepsis was performed using a povidone-iodine solution; then, a small incision was made on the anterolateral surface under an operating microscope. The tibia was sharply dissected proximally until the metaphyseal flair was identified and a small hole was created with a 27 G needle in the center of the bone. The needle was angled approximately 45° in the sagittal plane and advanced to penetrate the cortex. The needle was twisted gently to create a cortical window while leaving the posterior cortex intact to avoid fracture. K7M2 cells (5 × 10^5^ cells in10 µL of PBS) were then injected into the defect using a Hamilton pipette. The same procedure was performed in the case of the sham operation, but PBS was injected instead of tumor cells. The muscle was pulled back over the bone, and the skin was closed in a running 4-0 vicryl suture [37].

### 2.5. Measurement of the Mechanonociceptive Threshold

Weekly, we measured the mechanonociceptive threshold of the hind paws’ plantar surface using dynamic plantar aesthesiometry (Ugo Basile 37000). The mice were placed in transparent acrylic boxes with wire grid flooring. After acclimation, we mechanically stimulated the plantar surface with a straight metal filament, gradually increasing the upward force (up to a maximum of 10 g with a 4 s latency) until the animal withdrew its paw, indicating its mechanonociceptive threshold. Hyperalgesia was quantified as the percentage decrease compared to the baseline withdrawal thresholds [26,38].

### 2.6. Dynamic Weight Bearing

Articular nociception was assessed using a DWB apparatus (Bioseb, France). The apparatus comprises a small Plexiglas chamber (11.0 × 19.7 × 11.0 cm) equipped with floor sensors containing pressure transducers. The software associated with the system records, in grams, the average weight exerted by each limb on the floor, without any interference from the analyzer. During testing, the mouse was placed in the chamber and allowed to move freely for a period of 5 min. A camera positioned at the side of the enclosure aided in data analysis. All movements were recorded and validated by the experimenter to determine which paw corresponded to the set of pixels recognized by the sensors: right or left paw. The DWB software (Dynamic Weight Bearing 2.0, Bioseb, France) provided data on the weight (in grams) or area (in mm^2^) of the paw touching the floor, excluding the testicles and tail from the analysis. The animals underwent testing without prior adaptation, as exploratory movements were found to enhance data capture. The results were expressed as the percentage weight or area of the ipsilateral hind (right) and contralateral paw (left) [39].

### 2.7. Spontaneous Pain Behaviors

All mice were placed in suspended Plexiglas chambers with a wire grid floor. Guarding and flinching behaviors were measured during a 2 min observation period after a 30 min acclimation period. Flinching was defined as the lifting and rapid flexing of the ipsilateral hind paw not associated with walking or movement. Guarding was characterized by fully retracting the ipsilateral hind limb under the torso [40,41].

### 2.8. Measurements of Knee Diameter

The anteroposterior and mediolateral diameters of the knee joint of mice in each group were measured with a digital caliper (Mitutoyo). The change in knee diameter relative to the preoperative value was expressed as a percentage according to the following formula:$$Change\;in\;knee\;diameter=\left(\frac{Actual\;knee\;diameter\;\left(mm\right)}{Preop.\;\;knee\;diameter\;\left(mm\right)}\times 100\right)-100$$

### 2.9. Bone Structure Measurements

The right proximal tibia was repeatedly imaged (in the control and at 17 and 28 days) using in vivo micro-CT (SkyScan 1176, Bruker, Kontich, Belgium) with a voxel size of 17.5 μm. The imaging was conducted under ketamine-xylazine anesthesia, following previously established protocols (average anesthesia duration: 30–40 min, scan duration: 7–10 min). Bone structural changes were assessed using CT Analyser^®^ software (CT Analyzer 5.30 Software, OMICRON, Vienna, Austria), employing regions of interest (ROIs) of identical sizes around the tibia. ROIs of standard dimensions were delineated around the respective regions to calculate bone volume.

### 2.10. Glial Cell Immunohistochemistry and Evaluation

The animals were anesthetized with Euthasol and, after reaching the appropriate depth of anesthesia, transcardially perfused with 4% paraformaldehyde solution for immunohistochemistry. Following fixation, their spinal cords and brains were prepared and post-fixed for 12 h. For cryoprotection, we used a 30% sacharose solution. Immunohistochemistry was performed using the free-floating technique. Then, 30 µm sections were treated with a hydrogen peroxide solution for 20 min. The sections were incubated with diluted GFAP (monoclonal, 1:1000, Novocastra TM LeicaBiosystem) or Iba1 (polyclonal, 1:500, Wako, Richmond, VA, USA) antibodies for 48 h in 4 °C. Following the treatment with secondary biotin-conjugated anti-rabbit (Iba1) or anti-mice (GFAP) antibodies, we treated the sections with an avidin–biotin complex (ABC kit Vectastain). Dehydrated and coated sections were studied using a Nikon Eclipse Ni-E bright-field microscope. A quantitative analysis was performed using the Neurolucida system (Version 7, MicroBrightField, Williston, VT, USA). A previously described modified unbiased stereology protocol was used for the quantification of GFAP or Iba1 immunoreactive cells along the nociceptive pathways [42,43].

### 2.11. Statistical Analysis

All data were presented as means ± SEM and evaluated using the statistical software package GraphPad Prism v.8. (GraphPad Software, Inc., San Diego, CA, USA). Data distributions were analyzed with a Shapiro–Wilk normality test. Since all data were normally distributed, the results were statistically analyzed by repeated measures two-way analysis of variance (RM two-way ANOVA) followed by Tukey’s multiple comparison test. Since some values were missing due to mortality following the RTX pretreatment, these data were analyzed by fitting on a mixed model, mixed-effects analysis with Tukey’s multiple comparison test.

## 3. Results

### 3.1. Significant Mechanical Hyperalgesia, Weight Bearing Decrease and Spontaneous Pain-Related Behaviors in the Mouse Osteosarcoma Model after 10 Days Independent of Sex

The initial mechanonociceptive thresholds were 8.38 g and 8.17 g in the male and female BALB/c mice, respectively, and did not differ significantly. Significant mechanical hyperalgesia developed in the osteosarcoma cell-injected animals of both sexes (male: −21%, female −27%) from day 10 compared to the sham-operated groups, which further increased to 40–50% and remained during the 28-day experimental period (Figure 2A). Weight bearing on the ipsilateral hind limb compared to the contralateral side was significantly decreased (from 48–50% to 6–9%) in both sexes on days 14 and 28 in the osteosarcoma group, but not in the sham-operated animals (Figure 2B). Spontaneous pain-related behaviors like flinching and guarding of the osteosarcoma-affected limb were observed from day 14 in both sexes (Figure 2C,D). Despite the presence of the tumor, significant health impairments were not observed. Body weight measurements showed significant differences between sexes (Appendix A).

### 3.2. Osteosarcoma-Induced Tibia Diameter Increases after 21 Days in Both Sexes

The tibia diameter representing tumor growth gradually progressed from day 14 following intratibial injection of the cells in both sexes, reaching the maximum of 70–90% of the anteroposterior and mediolateral increase by day 27. No changes in the tibia diameter were observed in the sham-operated groups (Figure 3A,B).

### 3.3. Ostosarcoma-Induced Sex-Independent Osteoplastic Changes in the Tibia Microarchitecture

Micro-CT measurements revealed a bone volume increase due to osteoblastic tumor formation (male: from 5.6 to 7.5 mm^3^; female: from 4.3 to 6.2 mm^3^). The bone mass demonstrated by the bone volume/total volume ratios was significantly higher in the tibia of males compared to females, from 9% to 12% for males and from 7 to 10 for females, with sex differences (Figure 4A,B).

### 3.4. Significantly Increased Microglial and Astroglial Density in the Spinal Dorsal Horn in Response to Osteosarcoma Growth

The density of Iba1-positive cells related to microglial activation increased in the ipsilateral L4–L6 spinal dorsal horn in response to osteosarcoma development compared to the respective sham-operated groups on day 14 both in male (male: 8548.35 cell/mm^3^, female: 7390.88 cell/mm^3^) and female mice, but not on day 28, with no significant differences between the two sexes. Therefore, the male and female data were merged for further evaluation. Significant changes in Iba1 density were not detected in the PAG and SSC on day 14. On day 28, Iba1-positive cell densities were not altered in any investigated regions (Figure 5, Appendix A). GFAP immunopositivity related to astrocyte activation was augmented on day 28 following osteosarcoma induction compared to the sham-operated group, but not on day 14 (Figure 6, Appendix A).

### 3.5. Defunctionalization of the Capsaicin-Sensitive Sensory Fibers by RTX Pretreatment Does Not Influence Any Pain Parameters and Osteosarcoma Growth

Mechanical hyperalgesia (nociceptive threshold decrease; Figure 7A) developed in the osteosarcoma-injected animals, but there were no differences compared to the RTX-pretreated animals. In the case of weight bearing (Figure 7B), we can also see tumor formation, but not the effect of RTX desensitization. Spontaneous pain-related behaviors like flinching (Figure 7C) and guarding (Figure 7D) were also observed without any differences in pre- and non-pretreated animals. RTX desensitization did not change either the knee diameters (Figure 7E,F) or any bone structure parameters (Figure 8A,B).

## 4. Conclusions

This is the first comprehensive study to characterize the K7M2 orthotopic osteosarcoma mouse model with special emphasis on pain using in vivo functional and imaging techniques. We demonstrated the progression of this osteoblastic tumor followed by spontaneous and referred pain parameters, as well as neuroinflammation (microglial and astroglial activation) in the spinal dorsal horn, potentially leading to central pain sensitization. The capsaicin-sensitive subpopulation of nociceptive neurons does not seem to play a crucial role in osteosarcoma-related pain behaviors and cancer growth.

The first reported syngenic and orthotopic model using K7M2 osteosarcoma cells isolated from a spontaneously occurring tumor was performed in BALB/c mice. This is a valuable tool to investigate the pathophysiological mechanisms and novel treatment approaches in bone cancers and related pain [44]. Based on this paper, we investigated bone-cancer-related pain for 4 weeks, since after that time point, the metastatic burden rapidly increases, which is likely to influence the behavioral measurements [34].

Secondary mechanical hyperalgesia in the paw was the earliest detectable symptom, being apparent already on day 10, before any visible signs of tumor growth appear. It was followed by bone diameter (caliper) and volume (micro-CT) increases, spontaneous pain–related behaviors like weight-bearing reduction, flinching and guarding from day 17. According to our statement, neither the development of the tumor nor the pretreatment with RTX had any effect on the weight of the mice or their overall health status.

The mechanisms of CIBP are still unclear. The periosteum and bone marrow are densely innervated by sensory afferents and sympathetic nerves [45]. Ranges of inflammatory factors are rapidly secreted around the osteosarcoma cell injection site in the tibia, which creates an acidic local microenvironment leading to tissue destruction. These mechanisms result in peripheral sensitization at the level of the nociceptive nerve terminals and the cell bodies of the primary sensory neurons in the dorsal root ganglia (DRG), as well as central sensitization in the spinal dorsal horn and pain-related brain regions. This might explain the development of secondary hyperalgesia earlier than tumor growth and spontaneous pain. This finding is supported by the literature demonstrating that spontaneous pain fluctuates and develops after hyperalgesia in patients [8,46].

Micro-CT is a particularly useful tool for the sensitive detection of qualitative and quantitative microarchitectural changes in the bone induced by osteosarcoma [8]. In our study, the bone volume significantly increased by the end of the experiment in agreement with the knee diameter data. The initial, intact bone volume was significantly smaller in female mice at the beginning of the study, but not 4 weeks after osteosarcoma induction. This finding is in agreement with the literature showing sex differences in bone mineralization and vascularization, which correlates with age [47,48]. Normal bone remodeling and homeostasis are related to a balanced function between osteoclasts and osteoblasts. The invasion of cancer cells to the tumor environment disrupts this balance and can lead to both osteolysis and pathological osteogenesis depending on the type, origin and molecular characteristics of the cancer [8]. Osteosarcoma is an exception as it induces osteolysis and produces mineralized osteoid simultaneously [49]. This K7M2 osteosarcoma model is an osteoblastic tumor, as shown by the increased bone volume and mass, similarly to the human disease it models.

Previous studies highlighted the critical importance of astroglial and microglial activation in pain sensitization, maintenance and modulation [50,51]. Correlations of glial cell activation with hyperalgesia were demonstrated in different inflammatory pain conditions, such as the complete Freund’s adjuvant-induced rat model [52,53]. Our group previously showed significantly increased microglia-related (Iba1) but unchanged astrocyte marker levels (GFAP) in the dorsal horn of the lumbar spinal cord in the late stage of the K/BxN serum-transfer autoimmune arthritis model [28]. Another K/BxN model demonstrated persistent microglial activity at early and late time points, but astrocyte staining increased only during the early, inflammatory phase [54]. There are strong and complex interactions between microglia and astrocytes in neuroinflammatory mechanisms. We investigated glial cell activation in the spinal dorsal horn and pain-related brain regions 2 weeks (early phase) and 4 weeks (late phase, persistent pain) after osteosarcoma induction. In the L4–L6 spinal dorsal horn, Iba1 density significantly increased at 2 weeks and GFAP immunopositivity increased at the 4-week time point, indicating that increased mechanical hypersensitivity in this model is associated with transient microglial and persistent astrocyte activation in the CNS. Neither glia marker changed at neither time point in the PAG and the SSC.

After systemic defunctionalization of the capsaicin-sensitive peptidergic sensory fibers with RTX, we did not find any change either in osteosarcoma growth or the pain-related functional outcomes, suggesting that bone cancer pain is mediated predominantly by capsaicin-insensitive nociceptive afferents and presumably central sensitization mechanisms. In contrast to the present findings, we previously found increased tumor growth attributed to enhanced vascularity and permeability in the triple-negative 4T1 orthotopic mouse model of breast cancer after RTX desensitization [16]. This difference might be explained by distinct mechanisms involved in the complex sensory neuronal–vascular–immune interactions in the microenvironment of different types of cancers.

Severe bone cancer pain is due to multiple mechanisms with inflammatory, neuropathic and nociplastic components. Tumor cells can damage and destroy the distal processes of sensory fibers that normally innervate the bone. Activating transcription factor 3 (ATF3) was found to be overexpressed in mouse DRG in response to femoral osteosarcoma, suggesting the potential role of ATF3-expressing capsaicin-insensitive A β mechanosensitive neurons. Alternatively, it is possible that tumor cells induce a decrease in the expression of neuropeptides (like CGRP) or neurofilaments (such as NF200) in these sensory fibers [55,56].

Several studies showed the inhibitory effect of intrathecal RTX only desensitizing TRPV1-expressing cells in the central nervous system in serious pain conditions including bone cancer models [57,58,59]. Chronic administration of the TRPV1 antagonist JNJ-17203212 or disruption of the TRPV1 gene significantly attenuated, but not abolished, cancer-induced ongoing and movement-evoked nocifensive behaviors in mice [60]. Besides TRPV1, other receptors on the capsaicin-sensitive afferents, such as bradykinin, P2X3 and prostaglandin receptors, or acid sensing ion channel 3 and voltage-gated sodium channels may also be involved in this severe chronic pain state [61]. New research suggests that inflammatory and immune cells infiltrate damaged peripheral nerves, potentially releasing a range of cytokines and growth factors. These substances could play a role in both the onset and persistence of neuropathic pain [62].

A limitation of our study is that there are differences in TRPV1 expression patterns between mice and humans, potentially impacting pain perception and nociceptive signaling in the two species. In mice, TRPV1 is expressed in a subset of sensory neurons, around 22–38% at the mRNA level; however, in humans, TRPV1 is more widely expressed, around 70% [63,64]. Several previous datasets also showed differences between mice and human protein expression for nociceptor markers in the human DRG, dorsal horn, nociceptor populations and their central projection pathways [65,66,67]. These findings underscore the importance of verifying the target homology between rodent models and humans to ensure translational potential and therapeutic effectiveness. We are planning to propose validation targets based on RNA scope in situ hybridization to enhance the translational value. Non-neuronal TRPV1 could have a role in various cell types, like in neutrophils, macrophages and osteosarcoma cells, and might be involved in tumorigenesis and related pain [68,69,70]. However, our research group previously showed that non-neuronal TRPV1 expression was not altered after pretreatment with the ultra-potent capsaicin analog. Therefore, we suggest that RTX selectively defunctionalizes capsaicin-sensitive nociceptive fibers without affecting other TRPV1 expressing cells [71]. Therefore, understanding the differential effects of TRPV1 modulators on different cell types is crucial for developing targeted therapies for pain and tumorigenesis. Another limitation is the mouse origin of the osteosarcoma cell line; however, human cell lines can only be investigated in immunosuppressed animals, which greatly influences the tumor cell–neuron–vascular–immune interactions and the cancer microenvironment. Orthotopic mouse models most closely mimic the clinical course of cancer progression and do not rapidly induce distant metastases. It is thus concluded that osteosarcoma progression and pain are not affected by capsaicin-sensitive C/Aδ fibers, representing 50–70% of pain-sensing neurons, and that other nociceptive neuronal types are suggested to be involved. Our data suggest that in bone cancer-related severe chronic pain, an inhibiting neuroinflammatory mechanism might be a novel therapeutic approach. Therefore, exploring the molecular mechanism of astrocyte and microglia activation could help to identify pharmacological targets for developing effective analgesics in this condition. Special and/or single cell transcriptomics in pain-related brain regions could help to determine potential key mediators, the receptors, enzyme transporters and ion channels they target. Optogenetics, chemogenetics and pharmacogenetics, followed by genetic manipulations and pharmacological interventions, might be useful tools to validate the selective targets.

## Figures and Tables

**Figure 1 cancers-16-01788-f001:**
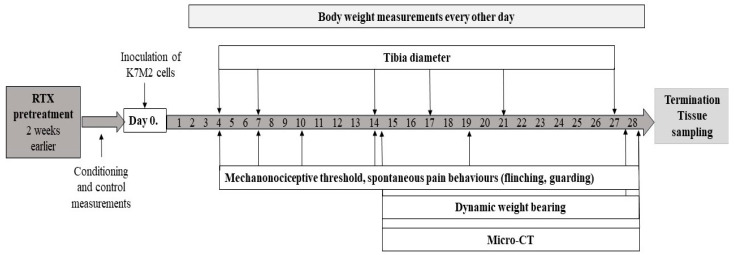
Summary of the design and timeline of the experiment. Following intratibial injection of the K7M2 cells, the mechanonociceptive threshold, tibia diameter, spontaneous pain behaviors, dynamic weight bearing and bone structural alterations (micro-CT) were measured at the indicated time points during the 28-day investigation period.

**Figure 2 cancers-16-01788-f002:**
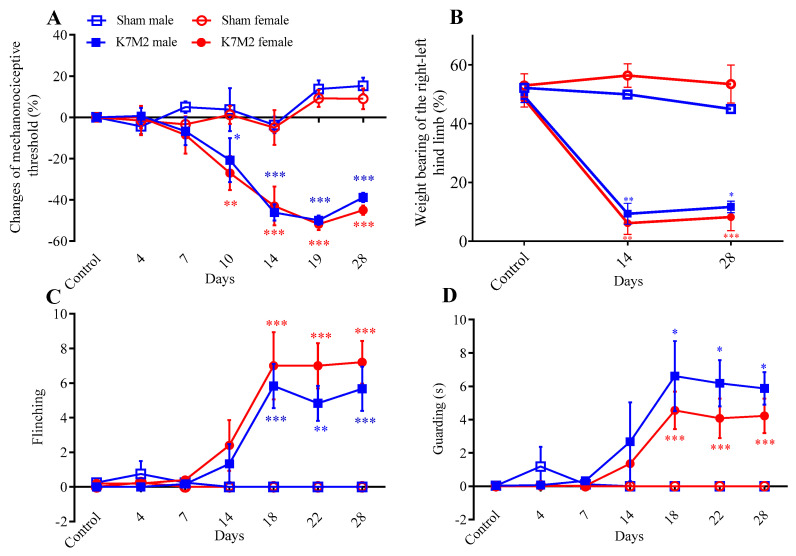
Osteosarcoma-induced hyperalgesia (**A**), decreased weight bearing of the right-left hind limbs (**B**), number of flinching (**C**) and time spent guarding (**D**) throughout the 28-day experimental period compared to the sham-operated groups. Data are shown as means ± S.E.M. of *n* = 4–5 mice/group, * *p* < 0.01, ** *p* < 0.001, *** *p* < 0.0001 vs. respective sham group (RM two-way ANOVA + Tukey’s multiple comparison test).

**Figure 3 cancers-16-01788-f003:**
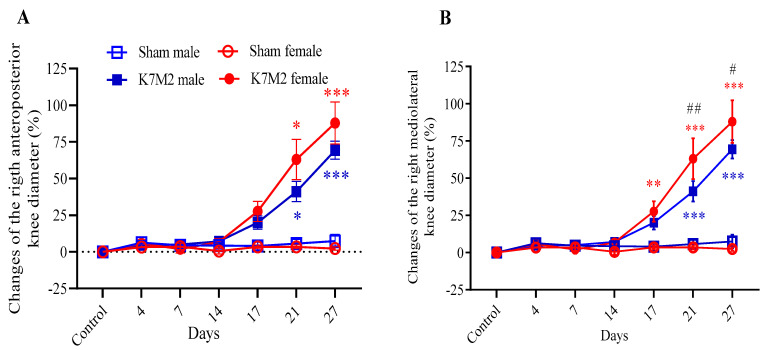
Percentage changes in the anteroposterior (**A**) and mediolateral knee diameters (**B**) of the affected limbs relative to the preoperative control values (*n* = 7–8; * *p* < 0.01, ** *p* < 0.001, *** *p* < 0.0001 vs. respective sham group, # *p* < 0.05, ## *p* < 0.01, vs. respective male. RM two-way ANOVA + Tukey’s multiple comparison test).

**Figure 4 cancers-16-01788-f004:**
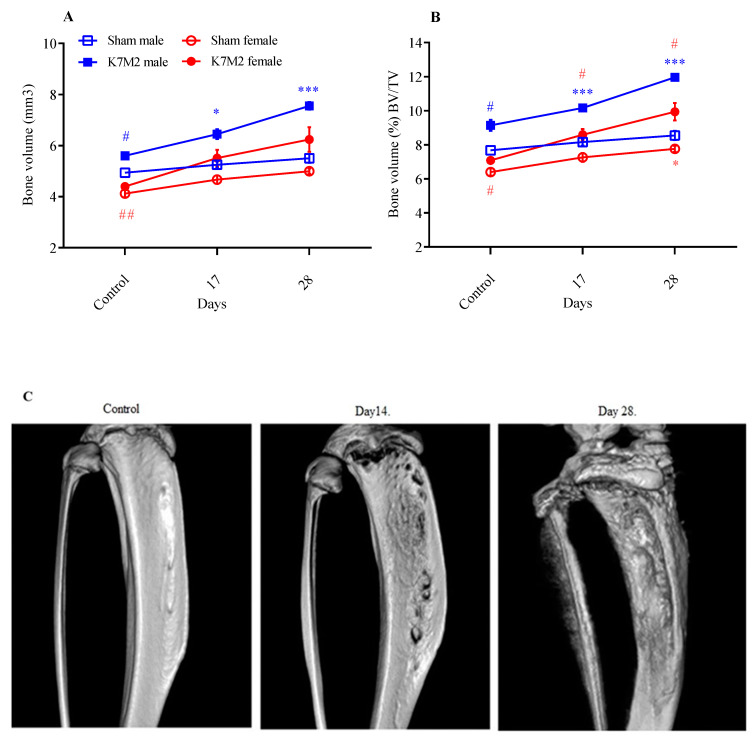
Significant increase in bone volume (**A**) and percentage of bone volume (BV/TV) (**B**) in the tibia at both the 17- and 28-day time points. * *p* < 0.01, *** *p* < 0.0001 vs. respective sham group, # *p* < 0.05, ## *p* > 0.001 vs. male; *n* = 4/group (RM two-way ANOVA, Tukey’s multiple comparison test) Representative micro-CT images of the tibia before and 14 and 28 days after osteosarcoma development (**C**).

**Figure 5 cancers-16-01788-f005:**
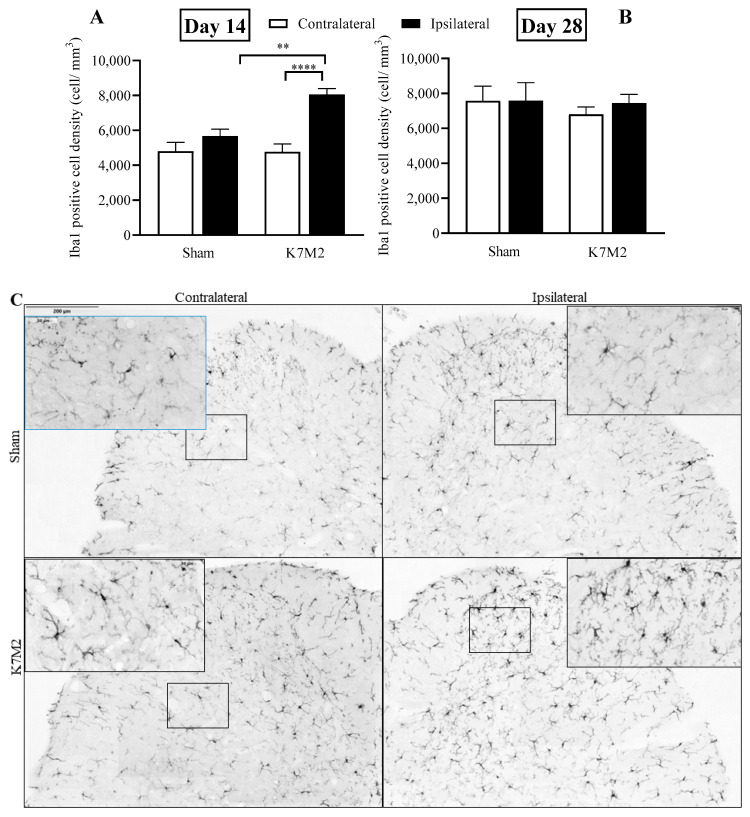
Iba1-positive microglia cells in the L4–6 spinal dorsal horn ipsilateral and contralateral to the tumor-injected side 14 (**A**) and 28 (**B**) days after osteosarcoma injection. Representative images showing Iba1 immunopositivity in the examined groups (**C**); ** *p* < 0.001, **** *p* < 0.0001 vs. respective sham, *n* = 6–9/group, males and females merged. One-way ANOVA followed by Bonferroni’s post hoc test.

**Figure 6 cancers-16-01788-f006:**
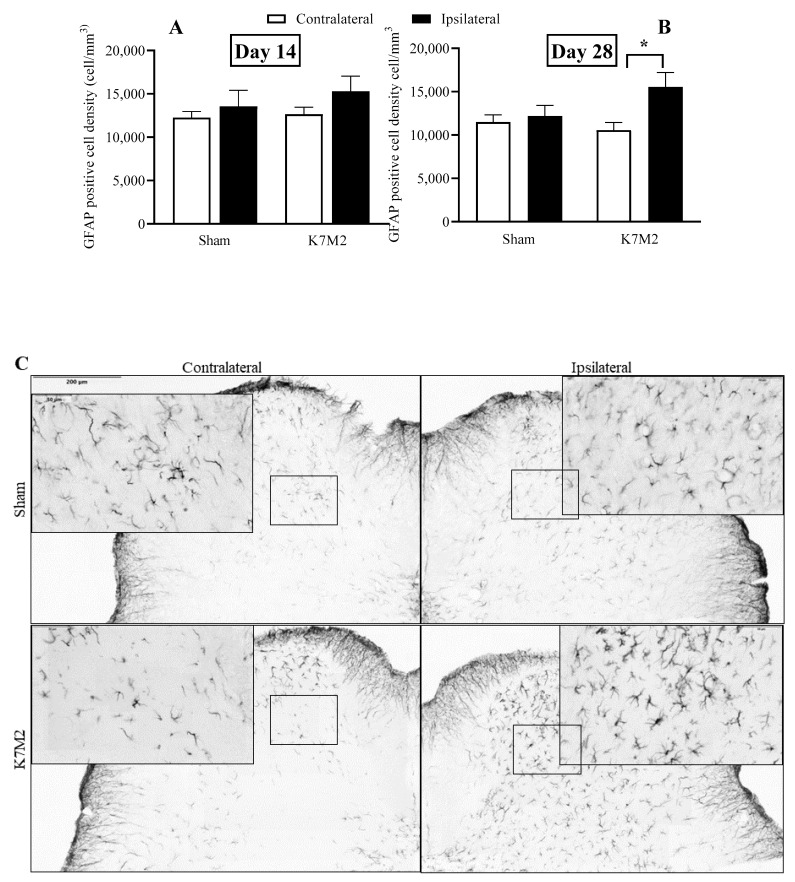
GFAP-positive astrocytes in the dorsal horn of the spinal cord (L4–6) ipsi- and contralateral sides on days 14 (**A**) and 28 (**B**) after tumor injection with a representative image of the spinal cord sections (**C**). * *p* < 0.05 vs. respective sham, *n* = 6–9/group, males and females merged. (One-way ANOVA followed by Bonferroni’s post hoc test).

**Figure 7 cancers-16-01788-f007:**
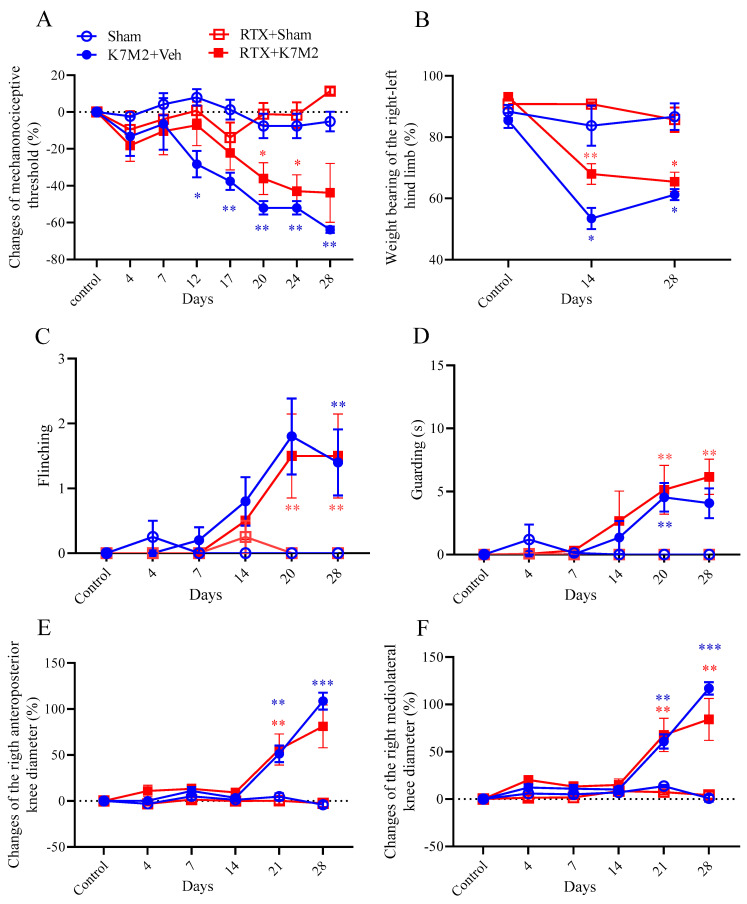
Changes in mechanonociceptive threshold (**A**), weight bearing (**B**), flinching (**C**) and guarding (**D**), and anteroposterior (**E**) and mediolateral (**F**) knee diameter throughout the 28-day experimental period. Data are shown as means ± S.E.M. of *n* = 7–10 mice/group, * *p* < 0.01, ** *p* < 0.001, *** *p* < 0.0001 vs. sham (RM two-way ANOVA and mixed-effects analyses following Tukey’s multiple comparison test).

**Figure 8 cancers-16-01788-f008:**
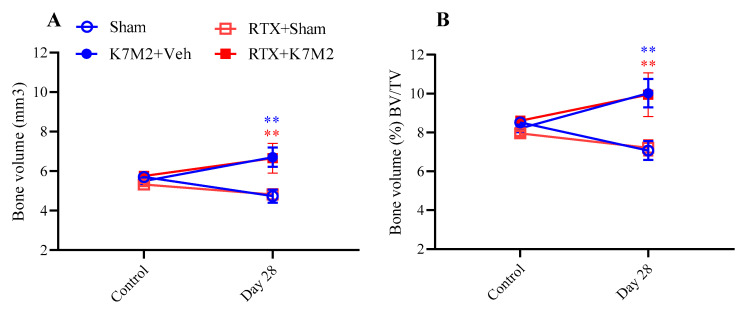
The increase in bone volume (**A**) and percentage of bone volume/total volume ratios (BV/TV) (**B**) of the tibia induced by K7M2 inoculation is not affected by RTX pretreatment on day 28, determined by micro-CT; ** *p* < 0.01 vs. respective sham; ** *p* < 0.01, vs. respective sham groups; *n* = 4/group (RM two-way ANOVA + Tukey’s multiple comparison test).

## Data Availability

The datasets presented in this article are available from the corresponding author on reasonable request.

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
