# Peer review of "Osteosarcoma-Induced Pain Is Mediated by Glial Cell Activation in the Spinal Dorsal Horn, but Not Capsaicin-Sensitive Nociceptive Neurons: A Complex Functional and Morphological Characterization in Mice"

_cancers, 2024, doi:10.3390/cancers16101788_

Round 1

Reviewer 1 Report

Comments and Suggestions for Authors

Dear authors,

Study represents bone cancer-related chronic pain and hyperalgesia are likely to be mediated by central sensitization involving neuroinflammation via glial cell activation in the spinal dorsal horn, but not the capsaicin-sensitive sensory neuronal system. it can be considered, after following incorporations. 

please add the novelty of your work in the abstract and introduction section.

you are suggested to add new references in the discussion section. 

you are suggested to enhance future perspectives of your work along with methods.

you are suggested to add schematic figures to represent the proposed mechanisms of your work. 

Reviewer 2 Report

Comments and Suggestions for Authors

Please check the text carefully. Several typos are present, especially in the introduction part.

Comments on the Quality of English Language

English can be improved with minor revisions

Author Response

Response to review report 2

Comments and Suggestions for Authors

Please check the text carefully. Several typos are present, especially in the introduction part.

Thank you for your comment, we have thoroughly reviewed it again and corrected the typing errors.

Comments on the Quality of English Language

English can be improved with minor revisions

Thank you for your recommendation, our manuscript has been carefully revised by a colleagues fluent in English writing.

Reviewer 3 Report

Comments and Suggestions for Authors

Comments on the Quality of English Language

minor editing

Author Response

Response to review report 3

  1. Overall, the paper by Bencze and colleagues is interesting. The introduction provides sufficient background with relevant current literature on the topic. Material and methods are adequately described. However, the abstract should be revised as per journal guidelines, and it should follow the style of structured abstracts: 1) Background: Place the question addressed in a broad context and highlight the purpose of the study; 2) Methods: Describe briefly the main methods or treatments applied. Include any relevant preregistration numbers, and species and strains of any animals used; 3) Results: Summarize the article's main findings; and 4) Conclusion: Indicate the main conclusions or interpretations. The abstract should be an objective representation of the article: it must not contain results which are not presented and substantiated in the main text and should not exaggerate the main conclusions.

We apologize for this mistake, which was due to technical problems during the uploading process. Several parts of the abstract were missing and the segmentation disappeared. It has now been corrected.

  1. Important and critical issue relates to immunoperoxidase images of Figure 5 and 6. Major issue related to Figure 5. Iba1 immunopositivity

The authors should carefully revise the images of the inserts because the immunopositive cells in two panels, which correspond to different experimental groups, look exactly the same though at a slightly different magnification. Again, for each panel, it should be clearly indicated with an outlined box/square in the main image the site from which the relative insert has been obtained

  1. Another major issue in Figure 6. GFAP immunopositivity. The authors should carefully revise the images of the inserts because there is a similarity between immunopositive cells from two different panels. Again, for each panel, it should be clearly indicated with an outlined box/square in the main image the site from which the relative insert has been obtained.

All this immunohistochemistry images have been carefully reviewed as suggested, the criticized panels have been modified, and higher magnification inserts have been shown to avoid potential inconveniences (modified figures 5., 6 ).

Reviewer 4 Report

Comments and Suggestions for Authors

The authors have carefully characterized osteosarcoma-induced bone cancer pain in BALB/c mice. The TRPV1 agonist resiniferatoxin (RTX) pre-treatment was used to ablate TRPV1 expressing sensory neurons. Obtained results are somewhat surprising as selective killing of TRPV1 sensory neurons did not affect spontaneous guarding pain, evoked mechanical pain responses and bone structural changes in both sexes. Transient microgliosis in the spinal dorsal horn as visualized by Iba1 immunopositivity was present at day 14 but not anymore at day 28 and increased GFAP immunopositivity was present at day 28. Obtained results support the idea that osteosarcoma-induced pain is initially peripherally driven but becomes centralized.

Can the authors provide reference showing that 70% of mouse sensory neurons express TRPV1 (lines 339-341). Typically, about 10-20% of freshly isolated mouse DRG neurons show functional responses to capsaicin stimulation in vitro. Further, recent RNA sequencing data shows similarly more modest TRPV1 expressing neuron population size in mice (PMID: 36690629). In human, TRPV1 expressing sensory neuron population is clearly larger ~60% (PMID: 33550628). How well can the results from mouse translate to human? Please clarify.

Histological data from tumor samples and its nerves would have been an interesting positive control to show successful ablation of TRPV1 peptidergic fibers. Further, it could have provided clues with regard to sensory neuron subtypes driving cancer pain. A previous study (PMID: 15817267) showed that bone tumor induces injury to peripheral nerves resulting in ATF3 upregulation. Could ATF3 (over)expressing A beta mechanosensory neuron population drive osteosarcoma-induced pain? Please discuss.

Osteosarcoma aggressiveness is reported to correlate with increased osteoclast activity (PMID: 19020756). It would have been interesting to see if serum bone turnover markers such as TRACP 5b and cathepsin K could correlate with pain intensity. Did you take any serum samples from osteosarcoma injected mice?

Is TRPV1 expressed in human immune cells such as monocytes, macrophages and osteoclasts that are associated with osteosarcoma tumors. Is there any data available concerning RTX-induced alteration in immune cell populations? Please comment.

In schematic figure body weight measurement were apparently performed every other day. However, I was not able to find the results. Body weight data would provide valuable information about overall health status of mice.

References 15 and 16 are duplicated, please remove extra reference.

Please add references 59-61 or is mismatch due to reference duplication? Please check references carefully.

Author Response

Response to review report 4

The authors have carefully characterized osteosarcoma-induced bone cancer pain in BALB/c mice. The TRPV1 agonist resiniferatoxin (RTX) pre-treatment was used to ablate TRPV1 expressing sensory neurons. Obtained results are somewhat surprising as selective killing of TRPV1 sensory neurons did not affect spontaneous guarding pain, evoked mechanical pain responses and bone structural changes in both sexes. Transient microgliosis in the spinal dorsal horn as visualized by Iba1 immunopositivity was present at day 14 but not anymore at day 28 and increased GFAP immunopositivity was present at day 28. Obtained results support the idea that osteosarcoma-induced pain is initially peripherally driven but becomes centralized.

  1. Can the authors provide reference showing that 70% of mouse sensory neurons express TRPV1 (lines 339-341). Typically, about 10-20% of freshly isolated mouse DRG neurons show functional responses to capsaicin stimulation in vitro. Further, recent RNA sequencing data shows similarly more modest TRPV1 expressing neuron population size in mice (PMID: 36690629). In human, TRPV1 expressing sensory neuron population is clearly larger ~60% (PMID: 33550628). How well can the results from mouse translate to human? Please clarify.

This statement regarding the mouse sensory neuronal population was incorrect, which has been corrected (page 13, lines 338-339). In the mouse TRPV1 mRNA is expressed in 22-38% of all sensory neurons, while this value is greater in humans (~60-70 %). Furthermore, differences have been described between mouse and human protein expressions for nociceptor markers in the DRG, spinal dorsal horn, and their central projections (Jung et al 2023, Shiers et al 2020, 2021). This information has been added to the discussion and the difficulties to translate the mouse results to humans was considered as a limitation of the study (page 14-15, lines 419-428).

  1. Histological data from tumor samples and its nerves would have been an interesting positive control to show successful ablation of TRPV1 peptidergic fibers. Further, it could have provided clues with regard to sensory neuron subtypes driving cancer pain. A previous study (PMID: 15817267) showed that bone tumor induces injury to peripheral nerves resulting in ATF3 upregulation. Could ATF3 (over)expressing A beta mechanosensory neuron population drive osteosarcoma-induced pain? Please discuss.

We thank the Reviewer for raising this idea, it is indeed a possible explanation for our findings. Although we have not investigated ATF3 expression and determining the neuronal subtypes in the osteosarcoma tissue was beyond the scope of our study, this point has been discussed and respective reference has been inserted as suggested (page 14, lines 400-407).

Severe bone cancer pain is due to multiple mechanisms with inflammatory, neuropathic and nociplastic components. The activating transcription factor 3 (ATF3) was found to be overexpressed in mouse DRG in response to femoral osteosarcoma suggesting potential role of ATF3- expressing capsaicin- insensitive A β mechanosensitive neurons (Peters et al 2005, Chen et al 2021).

  1. Osteosarcoma aggressiveness is reported to correlate with increased osteoclast activity (PMID: 19020756). It would have been interesting to see if serum bone turnover markers such as TRACP 5b and cathepsin K could correlate with pain intensity. Did you take any serum samples from osteosarcoma injected mice?

Thank you for this valuable comment. It would be indeed interesting to investigate TRAPCP5b in the cancer tissue as an indicator of active osteoclasts and serum cathepsin K to compare it with the results of human studies (Avnet et al 2008). However, in our experiment the animals were transcardially perfused with paraformaldehyde since the primary aim was to perform brain immunohistochemistry. Therefore, blood samples were not technically possible to be taken. This is planned to be addressed in future experimental series.

  1. Is TRPV1 expressed in human immune cells such as monocytes, macrophages and osteoclasts that are associated with osteosarcoma tumors? Is there any data available concerning RTX-induced alteration in immune cell populations? Please comment.

In the present study we did not investigate non-neuronal TRPV1 expression, it was beyond the scope. Potentially, non-neuronal TRPV1 on neutrophils and macrophages (Denda et al 2010, Stander et al 2004 ) and also on the osteosarcoma cells (Hudhud et al 2024) might be involved in tumorigenesis and related pain. This seems unlikely as we observe several neurochemical changes in both neuronal and non-neuronal cells within the ipsilateral DRG of tumor-bearing animals that are similar to alterations seen in other models of peripheral nerve injury. However, our research group earlier showed that non-neuronal TRPV1 expression was not altered after pretreatment with the ultra-potent capsaicin analog resiniferatoxin (Kun et al 2012) Therefore we suggest that RTX selectively defunctionalized capsaicin -sensitive nociceptive fibers without affecting other TRPV1 expressing cells.

  1. In schematic figure body weight measurement were apparently performed every other day. However, I was not able to find the results. Body weight data would provide valuable information about overall health status of mice.

The body weight data and their changes have been included in a supplementary figure (Supplementary figure 2.) demonstrating sex difference but no alterations in response to treatment and RTX-desensitization. This information has been in the result (line 251-253) and discussion section (line 351-353).

References 15 and 16 are duplicated, please remove extra reference. Please add references 59-61 or is mismatch due to reference duplication? Please check references carefully.

Thank you for drawing our attention to reference mismatch, we corrected and checked is again.